# Estimation of true dates of various flowering stages at a centennial scale by applying a Bayesian statistical state space model

Nagai Shin[1]☯*, Hakuryu Fujiwara[2]☯, Shinjiro Sugiyama[3], Hiroshi Morimoto[4], Taku M. Saitoh[5]

1 Research Institute for Global Change, Japan Agency for Marine-Earth Science and Technology, Yokosuka, Japan, 2 Amateur Photographer, Motosu City, Japan, 3 Neo Community Center, Motosu City, Japan, 4 Emeritus Professor, Nagoya University, Nagoya, Japan, 5 Center for Environmental and Societal Sustainability, Gifu University, Gifu, Japan

☯ These authors contributed equally to this work.
* nagais@jamstec.go.jp

**Data Availability Statement:** A part of data that support the findings of this study are available from the web site in Motosu City. As for other data, we want to publish the part of observation records

## Abstract

Evaluation of long-term detailed cherry flowering phenology is required for a deep understanding of the sensitivity of spring phenology to climate change and its effect on cultural ecosystem services. Neodani Usuzumi-zakura (*Cerasus itosakura*) is a famous cherry tree in Gifu, Japan. On the basis of detailed decadal flowering phenology information published on the World Wide Web, we estimated the probability distributions of the year-to-year variability of the true dates of first flowering (FFL), first full bloom (FFB), last full bloom (LFB), and last flowering (LFL) from 1924 to 2024 by applying a Bayesian statistical state space model explained by air temperature data. We verified the estimated values against flowering phenology records of the tree from the literature and a private collection. The true dates of FFL and FFB could be explained by means of daily minimum air temperature from 1 December to 28/29 February and that of daily mean air temperature from 1 to 31 March, and those of LFB and LFL by means of daily mean air temperature from 1 to 10 April. Results were similar when we used air temperature data recorded at weather stations both 1 km and 29 km from the tree. These results indicated that our proposed Bayesian statistical state space model can estimate cherry flowering phenology that takes into account centennial-scale air temperature data recorded at a nearby weather station with a coarse temporal resolution.

## Introduction

Long-term records or prediction of cherry flowering phenology play an important role in the evaluation of changes in the sensitivity of spring plant phenology (i.e., flowering and leaf flushing) to climate change and its effect on cultural ecosystem services. As many deciduous trees flush new leaves at the same time as or after cherry flowering in Japan [1], cherry flowering phenology can be used as a representative spring plant phenology. Japan celebrates a traditional culture of *Hanami*, or cherry-blossom viewing [2]. For this reason, long-term records of

(without the copyright issue). Therefore, we have updated the supplementary file (csv). Data are also available from the authors upon reasonable request and with permission of Motosu City (a part of data observed by Motosu City).

**Funding:** NS was supported by a KAKENHI grant (JSPS 21H05178 and JP24K21357) from the Japan Society for the Promotion of Science. https://www.jsps.go.jp/english/e-grants/index.html This grant supported our activities regarding data collection and preparation of the manuscript.

**Competing interests:** The authors have declared that no competing interests exist.

*Hanami* festivals in the Imperial Palace since the 9th century, which can be used as proxy data for cherry flowering phenology [3], can be retrieved. Old diaries written by a citizen in the 19th century (middle Edo era) also allowed us to retrieve continuous records of cherry flowering phenology [4]. Since the early 20th century (late Meiji era), weather stations throughout Japan have collected phenological observations of various animals and plants. The records of cherry flowering date back to 1903 in Miyagi Prefecture [5,6] and to 1909 in Kyoto [7]. The Japanese Meteorological Agency (JMA) standardized phenological observations in 1953 [8], but we can retrieve the records of cherry flowering phenology (mainly Yoshino cherry, *Cerasus × yedoensis*) before then at multiple weather stations in Japan. Analysis of these data found that: (1) the dates of first flowering (FFL) throughout Japan advanced by 1.2 days per decade from 1953 to 2022 [9]; and (2) the correlation between dates of FFL and latitude decreased from 1953 to 2020 owing to delayed release of endodormancy in low latitude regions where annual mean air temperature is high [10].

Previous studies based on the long-term continuous records of cherry flowering phenology at multiple weather stations or popular cherry blossom viewing sites in Japan predicted the dates of FFL and first full bloom (FFB) by semi-empirical statistical phenology models (so-called "degree-day models" [11−13]) and machine learning algorithms (e.g., self-organizing maps [10], random forests, artificial neural networks, and gradient-boosting decision trees [14]). However, these models and algorithms have three issues. First, no model or algorithm can predict the dates of the last full bloom (LFB) and last flowering (LFL). The periods from FFL to FFB or from FFB to LFB differ among years owing to the day-to-day variability of temperature after FFL. To accurately evaluate the sensitivity of spring plant phenology to climate change [15] and thus of cultural ecosystem services generated by cherry flowering [12,13,16,17], long-term continuous records or predicted dates of LFB and LFL as well as of FFL and FFB are required. Second, in a semi-empirical statistical model [11−13], it is required to optimize each parameter by using daily or hourly air temperature data. The accuracy of a general semi-empirical statistical model applied to all observation sites in Japan was remarkably poor in the south, where the annual mean air temperature is high [11]. To improve the accuracy of semi-empirical statistical models, optimization of parameters at each observation site is required, but calculation is labor-intensive. Third, although websites of weather service companies, governmental and municipal offices, and tourism associations provide dates of flowering phenology at a daily time step at famous cherry blossom viewing spots in Japan (e.g., "tenki.jp" [18], "Weather News" [19], "Neodani Usuzumi-zakura" [20], "Miharu Taki-zakura" [21]), many of those popular cherry blossom viewing sites are distant from weather stations. To develop semi-empirical statistical models for these sites, correction of air temperature data observed at a nearby weather station (e.g., by applying the temperature lapse rate with altitude) is required, but the corrected data include uncertainty due to microclimate effects.

To resolve these difficulties, we propose a Bayesian statistical state space model that takes into account temperature data recorded at nearby weather stations. A major advantage of this model is that its estimates of the actual state (i.e., true dates of FFL or FFB) are output as a probability distribution by updating the prior distribution to the posterior distribution with new observed data under conditions of ambiguity, as explained in "Material and methods". We estimated the year-to-year variability of the true dates of FFL, FFB, LFB, and LFL of Neodani Usuzumi-zakura in Gifu, Japan, at a centennial time scale by applying a Bayesian statistical state space model. Many conventional studies have tried to predict the unique flowering dates and to minimize error. However, long-term prediction tends to be unstable. If the method is based on literal diaries, then the prediction becomes more unreliable. To bring this uncertainty under control, we predicted the distribution of flowering dates rather than unique flowering dates. The Bayesian method then enables us to evaluate the flowering dates from the

probability distribution. In other words, using the probability distribution, we can estimate quantitatively the uncertainty of prediction. The concept of our proposed model based on the Bayesian statistics is definitely different from that of conventional phenology models based on the traditional statistics [3,10,12,13]. The aims of this study were (1) to develop a cherry flowering phenology model to estimate the dates of FFL, FFB, LFB, and LFL from centennial-scale coarse (i.e., monthly or weekly) temperature data recorded at a nearby weather station and (2) to discuss the utility and uncertainty of this proposed.

## Materials and methods

### Records of flowering phenology

Our target was the ~1500-year-old Neodani Usuzumi-zakura (*Cerasus itosakura*), in Motosu City, Gifu Prefecture, Japan (35°37′55.9″N, 136°36′31.7″E; 200 m a.s.l.) [20]. The Motosu City Office website has published the dates of FFL, 20%−30% and 50% flowering, FFB, and LFB since 1989, the dates of first scattering since 2001, and the dates of LFL since 2002. It provides daily flowering phenology information during the flowering season, and a live feed of the tree [20]. We used the records of the dates of FFL, FFB, LFB, and LFL.

To validate the accuracy of model estimation before 1989, we retrieved records of cherry flowering phenology from the literature in the National Diet Library, Gifu City Central Library, and Motosu City Library and a private collection (so-called "historical dark data" [22,23]). We retrieved the records of 17 years of FFL dates and 13 years of FFB dates from the literature [24–32], a photograph in 1930 [33], and unpublished data of Motosu City from 1981 to 1988 (Motosu City, unpublished). The JMA literature refers to the tree as *C. itosakura* except in 1937. Although it is unclear whether these records (except in 1937) indicate Neodani Usuzumi-zakura, we used them. A photograph might be considered to be taken from FFB to LFB dates. We retrieved records of 48 years of FFL dates since 1977 and 64 years of FFB dates since 1955 from the private collection of Hakuryu Fujiwara, an amateur photographer and one of coauthors of this study. His definition of full bloom was that there were no unopened buds on the canopy as viewed through a telescope. Few to no flowers bloomed in several years from 1955 to around 1980 because the tree's health was compromised, but after root surgery, the tree's vigor recovered. Part of his data (1968–1990) is published [34]. Observation records, which were converted to the day of year (DOY) for analysis, in literature, photographs, and the private collection of Hakuryu Fujiwara are listed in the (S1 File).

### Air temperature data

We used air temperature data since 1924 recorded at the Gifu weather station (35°24′02.2″N, 136°45′45.1″E, 12.7 m a.s.l.), about 29 km southeast of the Tarumi weather station (35°38′20.8″N, 136°36′11.8″E; 190 m a.s.l.), and since November 1978 recorded at the Tarumi weather station, about 1 km northeast of Neodani Usuzumi-zakura. The annual mean temperature (1991–2020) differed between Gifu and Tarumi by 3.4°C [35].

The dates of FFL, FFB, and LFB from 1989 to 2024 and that of LFL from 2002 to 2024 had means (and SDs) of (FFL) DOY 92.0 (6.2), (FFB) 96.9 (5.9), (LFB) 102.1 (5.6), and (LFL) 106.3 (6.0) (Fig 1). On this basis, we used the average daily minimum air temperature from 1 December to 28/29 February to account for the chilling requirement for release from endodormancy, the average daily mean air temperature in March to account for the heat requirement for the growth of flower buds, and the average daily mean air temperature from 1 to 10 April to account for the heat requirement for further growth of flower and leaf buds [11–13,36].

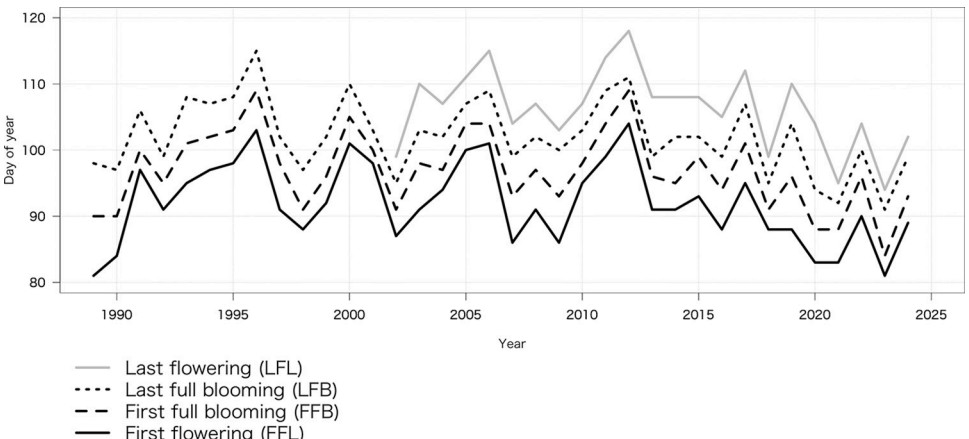

**Fig 1. Flowering phenology dates of Neodani Usuzumi-zakura from 1989 to 2024 published on the Motosu City Web site.**

## Flowering phenology model

Cherry trees form flower buds during summer, which then enter endodormancy. Exposure to winter cold releases the endodormancy. The flower buds then develop as heat increases [11−13,17,36]. To explain these processes, we constructed two varying coefficient regression models for the dates of FFL or FFB and of LFB or LFL, respectively. To validate the model estimation based on air temperature data recorded at the Gifu weather station during the period without data at the Tarumi weather station (1924–1978), we estimated the dates of FFL, FFB, LFB, and LFL by using air temperature data in Gifu from 1924 to 2024 and in Tarumi from 1979 to 2024 and compared the results.

The proposed Bayesian statistical state space model has three merits. First, its estimates of the actual state are output as a probability distribution and are explained as the essential variability once noise such as observation errors is removed. This merit reduces the uncertainty of observed data caused by qualitative visual inspection, individual differences among trees (i.e., resolving the issue of the representativeness of sampling data), and effects of microclimate. Second, it is not required to optimize each parameter, as in a semi-empirical statistical model [11−13]. Third, cherry flowering phenology can be estimated by using explanatory factors with a coarse temporal resolution (i.e., monthly or weekly air temperatures). These two merits reduce the complicated processes and labor-intensive calculations. Two previous studies have applied the Bayesian inference to phenology models but differed from the current study that made use of a state space model to estimate true dates [17,37].

## Estimation of the dates of FFL and FFB

To estimate dates of FFL and FFB, as explanatory variables we set the average daily minimum air temperature from 1 December to 28/29 February and monthly mean air temperature in March. During this period, the true date of FFL or FFB $\alpha1_t$ is explained by a general linearized model with $ex1_t$ (average daily minimum air temperature from 1 December to 28/29 February) as explanatory variable 1, and $ex2_t$ (monthly mean temperature in March) as explanatory variable 2 in year $t$ (Eq 4). We assumed that coefficients $\beta1_t$ of $ex1_t$ and $\gamma1_t$ of $ex2_t$ and variable $\mu1_t$ as the intercept vary according to the random walk model (i.e., state space equations: Eqs 1−3). On the other hand, $y1_t$ as the observed value of FFL and FFB dates is explained by the sum of $\alpha1_t$ (actual state in year $t$; i.e., true dates of FFL or FFB) and white noise (i.e., observation

equation: Eq 5):

$$\mu1_t = \mu1_{t-1} + \omega1_t, \omega1_t \sim Normal(0, \sigma^2_{\omega1}) \tag{1}$$

$$\beta1_t = \beta1_{t-1} + \tau1_t, \tau1_t \sim Normal(0, \sigma^2_{\tau1}) \tag{2}$$

$$\gamma1_t = \gamma1_{t-1} + \upsilon1_t, \upsilon1_t \sim Normal(0, \sigma^2_{\upsilon1}) \tag{3}$$

$$\alpha1_t = \mu1_t + \beta1_t \cdot ex1_t, +\gamma1_t \cdot ex2_t \tag{4}$$

$$y1_t = \alpha1_t + \varphi1_t, \varphi1_t \sim Normal(0, \sigma^2_{\varphi1}) \tag{5}$$

where $\omega1_t \sim Normal(0, \sigma^2_{\omega1})$ follows a normal distribution with a mean $\omega1_t$ of 0 and a variance of $\sigma^2_{\omega1}$. Terms $\sigma_{\omega1}$, $\sigma_{\tau1}$, and $\sigma_{\upsilon1}$ are the SDs (standard deviations) of the process errors, and $\sigma_{\varphi1}$ is the SD of the observation error.

## Estimation of the dates of LFB and LFL

To estimate dates of LFB and LFL, as an explanatory variable we set the average daily mean air temperature from 1 to 10 April. During this period, the true date of LFB or LFL, $\alpha2_t$, is explained by a general linearized model with $ex3_t$ (average daily mean air temperature from 1 to 10 April) as explanatory variable 3 in year $t$ (Eq 8). We assumed that coefficient $\beta2_t$ of $ex3_t$ and variable $\mu2_t$ as the intercept varied according to a random walk model (i.e., state space equations: Eqs 6 and 7). On the other hand, $y2_t$ as the observed value of LFB and LFL dates is explained by the sum of $\alpha2_t$ (actual state in year $t$; i.e., true date of LFB or LFL) and white noise (i.e., observation equation: Eq 9):

$$\mu2_t = \mu2_{t-1} + \omega2_t, \omega2_t \sim Normal(0, \sigma^2_{\omega2}) \tag{6}$$

$$\beta2_t = \beta2_{t-1} + \tau2_t, \tau2_t \sim Normal(0, \sigma^2_{\tau2}) \tag{7}$$

$$\alpha2_t = \mu2_t + \beta2_t \cdot ex3_t \tag{8}$$

$$y2_t = \alpha2_t + \varphi2_t, \varphi2_t \sim Normal(0, \sigma^2_{\varphi2}) \tag{9}$$

where $\omega2_t \sim Normal(0, \sigma^2_{\omega2})$ follows a normal distribution with a mean $\omega2_t$ of 0 and a variance of $\sigma^2_{\omega2}$. Terms $\sigma_{\omega2}$ and $\sigma_{\tau2}$ are the SDs of the process errors, and $\sigma_{\varphi2}$ is the SD of the observation error.

When we applied the state space model (Eqs 1–9) to actual data, we updated the prior distribution to the posterior distribution through the use of Bayesian inference. For example, the Bayesian update rule of the observation equation (Eq 5) is explained as:

$$P(\alpha1_t|y1_t) \propto P(y1_t|\alpha1_t) \times P(\alpha1_t) \tag{10}$$

where $P(\alpha1_t|y1_t)$, $P(y1_t|\alpha1_t)$, and $P(\alpha1_t)$ are the posterior distribution, likelihood, and prior distribution, respectively. We set the prior distribution $\alpha1_0$, $\alpha2_0$, $\beta1_0$, and $\beta2_0$ as uniform (i.e., non-informative). We used Markov chain Monte Carlo methods for generation of the random numbers following the posterior distribution. We implemented it by embedding RStan packages [38] in R software [39,40].

We set the number of chains (number of times to generate a set of random numbers) to 4, the number of repetitions of random number generation to 100 000, 150 000, or 200 000, the

burn-in period (initial data discarded) to 20 000, 30 000, or 50 000, and "thin" to 2. The model was run in R v. 4.2.2 and RStan v. 2.21.8 software on the RStudio desktop v. *R*2022.02.3 build 492 platform [41]. We modified the publicly available code of Baba (2019) [40,42].

## Results

### Estimation of flowering phenology from air temperature data recorded at the Gifu weather station

The estimated true dates of FFL, FFB, LFB, and LFL from 1924 to 2024 based on air temperature data recorded at the Gifu weather station ($\alpha 1_t$ and $\alpha 2_t$ in Eqs 4 and 8) are shown in Fig 2. Despite annual fluctuations, the median of the estimated true date of FFL advanced by 1.877 days/decade ($R^2 = 0.45$, $P < 0.001$), that of FFB by 1.901 days/decade ($R^2 = 0.48$, $P < 0.001$), that of LFB by 0.750 days/decade ($R^2 = 0.29$, $P < 0.001$), and that of LFL by 0.953 days/decade ($R^2 = 0.30$, $P < 0.001$).

The median of the estimated variables $\mu 1_t$ and $\mu 2_t$ as the intercept, coefficient $\beta 1_t$ of $ex1_t$ as explanatory variable 1, coefficient $\gamma 1_t$ of $ex2_t$ as explanatory variable 2, and coefficient $\beta 2_t$ of $ex3_t$ as explanatory variable 3 ($\mu 1_t$, $\beta 1_t$, and $\gamma 1_t$ in Eqs 1–3, and $\mu 2_t$ and $\beta 2_t$ in Eqs 6 and 7) are summarized in Table 1. Coefficients $\beta 1_t$ and $\gamma 1_t$ negatively affected the estimated true dates of FFL and FFB. For example, a 1˚C increase of the average daily minimum air temperature from 1 December to 28/29 February advanced FFL by 3.09 days, and a 1˚C increase of the monthly mean temperature in March advanced FFL by 3.75 days. Similarly, $\beta 2_t$ negatively affected the estimated true dates of LFB and LFL.

Most of the observed values ($y1_t$ and $y2_t$ in Eqs 5 and 9) of FFL, FFB, LFB, and LFL lay within the 95% credible interval of the estimated true dates from 1924 to 2024 ($\alpha 1_t$ and $\alpha 2_t$ in Eqs 4 and 8). Almost no discrepancies were found between $\alpha 1_t$ (median of estimated true date) and observed values of FFL and FFB before 1940. In contrast, $\alpha 1_t$ (the year-to-year variability of estimated true date) was not inclined to match the dates of FFB from 1955 to 1975 recorded by Hakuryu Fujiwara (during the period when the tree's vigor was reduced).

### Estimation of flowering phenology from air temperature data recorded at the Tarumi weather station

The estimated true dates of FFL, FFB, LFB, and LFL from 1979 to 2024 based on air temperature data recorded at the Tarumi weather station ($\alpha 1_t$ and $\alpha 2_t$ in Eqs 4 and 8) are shown in Fig 4. Despite annual fluctuations, the median of the estimated true date of FFL advanced by 1.823 days/decade ($R^2 = 0.18$, $P < 0.01$), that of FFB by 1.921 days/decade ($R^2 = 0.20$, $P < 0.01$), that of LFB by 0.109 days/decade ($R^2 = 0.14$, $P < 0.05$), and that of LFL by 0.116 days/decade ($R^2 = 0.11$, $P < 0.05$).

The medians of the estimated variables $\mu 1_t$ and $\mu 2_t$ and coefficients $\beta 1_t$, $\gamma 1_t$, and $\beta 2_t$ are summarized in Table 2. Coefficients $\beta 1_t$ and $\gamma 1_t$ negatively affected the estimated true dates of FFL and FFB. Similarly, $\beta 2_t$ negatively affected the estimated true dates of LFB and LFL.

Most of the observed values ($y1_t$ and $y2_t$ in Eqs 5 and 9) of FFL, FFB, LFB, and LFL lay within the 95% credible interval of the estimated true dates.

## Discussion

Applying a Bayesian statistical state space model allowed us to estimate the year-to-year variability of the true dates of FFL, FFB, LFB, and LFL at a centennial scale, covering the period when there were no records of the flowering phenology of Neodani Usuzumi-zakura (Fig 2). In particular, we could estimate the true dates at almost the same accuracy by models

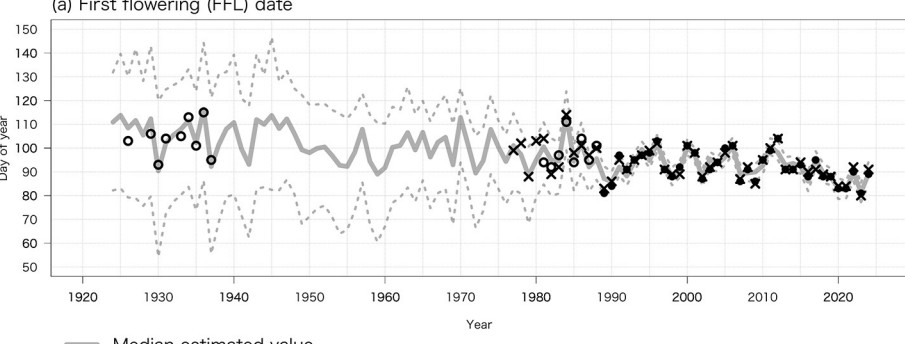

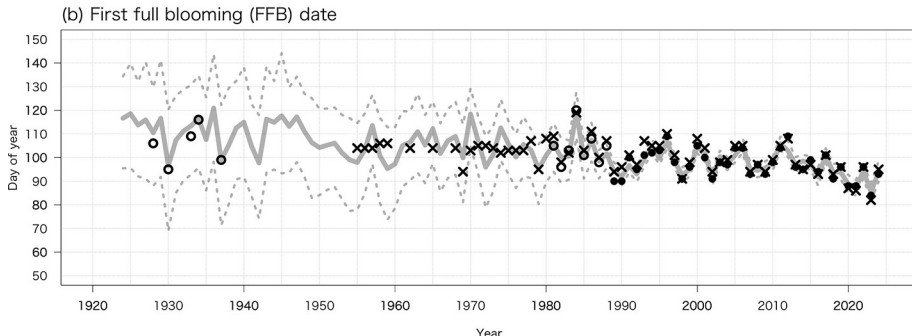

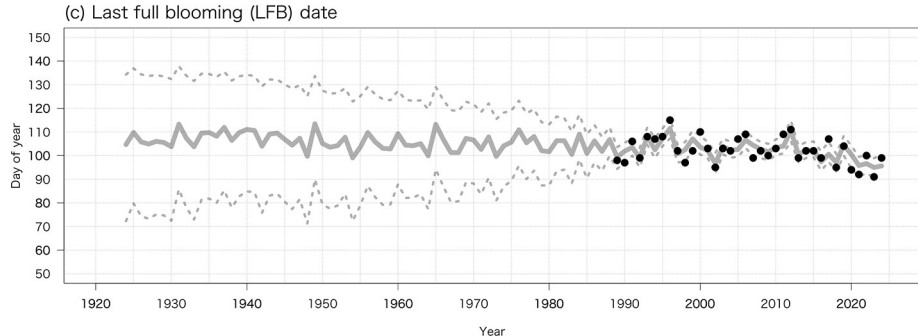

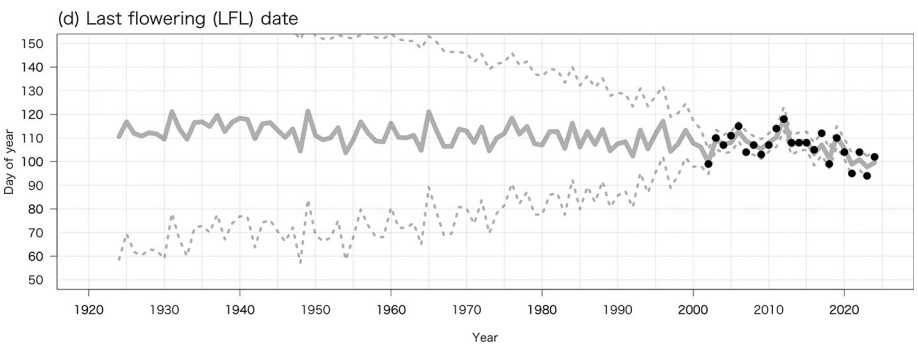

**Fig 2.** Relationship between probability distributions of estimated true dates of (a) FFL, (b) FFB, (c) LFB, and (d) LFL from 1924 to 2024 ($\alpha 1_t$ and $\alpha 2_t$ in Eqs 4 & 8) by the proposed Bayesian statistical state space model based on temperature data at Gifu weather station and observed values in records from Motosu City, the literature, and a private collection ($y1_t$ and $y2_t$ in Eqs 5 & 9).

**Table 1. Medians of estimated variables μ1_t and μ2_t as the intercept, coefficient β1_t of *ex*1_t as explanatory variable 1, coefficient γ1_t of *ex*2_t as explanatory variable 2, and coefficient β2_t of *ex*3_t as explanatory variable 3 based on air temperature data recorded at the Gifu weather station.** As an example, in the case of the estimated model for the true date of FFL, we show the probability distributions of estimated variable μ1_t and coefficients β1_t and γ1_t in Fig 3.

| Phenology | Parameter | Average | SD |
|---|---|---|---|
| True FFL date | μ1_t | 125.31 | 0.35 |
| | β1_t | −3.09 | 0.03 |
| | γ1_t | −3.75 | 0.47 |
| True FFB date | μ1_t | 131.92 | 0.23 |
| | β1_t | −3.31 | 0.02 |
| | γ1_t | −3.19 | 0.38 |
| True LFB date | μ2_t | 130.46 | 0.43 |
| | β2_t | −2.20 | 0.03 |
| True LFL date | μ2_t | 142.30 | 0.39 |
| | β2_t | −2.72 | 0.04 |

FFL, first flowering; FFB, first full bloom; LFB, last full bloom; LFL, last flowering.

explained by air temperature data recorded at weather stations both 1 km and 29 km from Neodani Usuzumi-zakura with a coarse temporal resolution (Figs 2 and 4). The mean daily air temperatures from 1 January to 31 December from 1991 to 2020 observed at Gifu and Tarumi had a significant positive correlation ($r = 1.0$, $P < 0.001$, degree of freedom 363 [43]). Despite the effect of microclimatology on air temperature, this means that there were no extreme discrepancies in patterns of day-to-day variability of air temperature between the weather stations. For this reason, the values of the coefficients β1_t, γ1_t, and β2_t (in Eqs 4 and 8) differed between models (Tables 1 and 2). Interestingly, the coefficient β1_t of the average daily minimum air temperature from 1 December to 28/29 February, which accounts for the chilling requirement for release from endodormancy, and the coefficient γ1_t of the average daily mean air temperature in March, which accounts for the heat requirement for the growth of flower buds, negatively affected the true dates of FFL and FFB (Tables 1 and 2). If sufficient exposure to winter cold is not obtained, the timing of release from endodormancy and subsequent flowering will be delayed [10,11]. However, despite the current warming climate, the average daily minimum air temperature from 1 December to 28/29 February at the Gifu and Tarumi weather stations may allow sufficient chilling for release from endodormancy, and subsequent warmer-than-average conditions during January or February would accelerate the growth of flower buds.

The probability distributions of the coefficients β1_t, γ1_t, and β2_t (in Eqs 4 and 8) were updated through the Bayesian update of the observation equation (Eqs 5 and 9) and state space equations (Eqs 1, 2, 6 and 7) when we obtained the observed values of FFL and FFB $y1_t$ or of LFB and LFL $y2_t$ (Fig 3). The values of FFL, FFB, and LFB were first observed in 1989 and that of LFL in 2002 (Fig 1). The median of estimated coefficients β1_t, γ1_t, and β2_t were almost constant before those years, while their credible intervals gradually widened into the past (Fig 3). However, if we obtain observed dates of FFL, FFB, LFB, and LFL $y1_t$ or $y2_t$ before 1989 or 2002, we can improve the accuracy of the estimated true dates; that is, the width of the 95% credible interval of the estimated true dates will be narrowed. This is another advantage of the proposed Bayesian statistical state space model. As an example, we estimated the true date of FFB from 1924 to 2024 both from flowering phenology records of the tree from the literature and a private collection before 1989 and from observed data published on the website since 1989 as the observed value of FFB $y1_t$ (Fig 5). Compared with the values of the true date of FFB

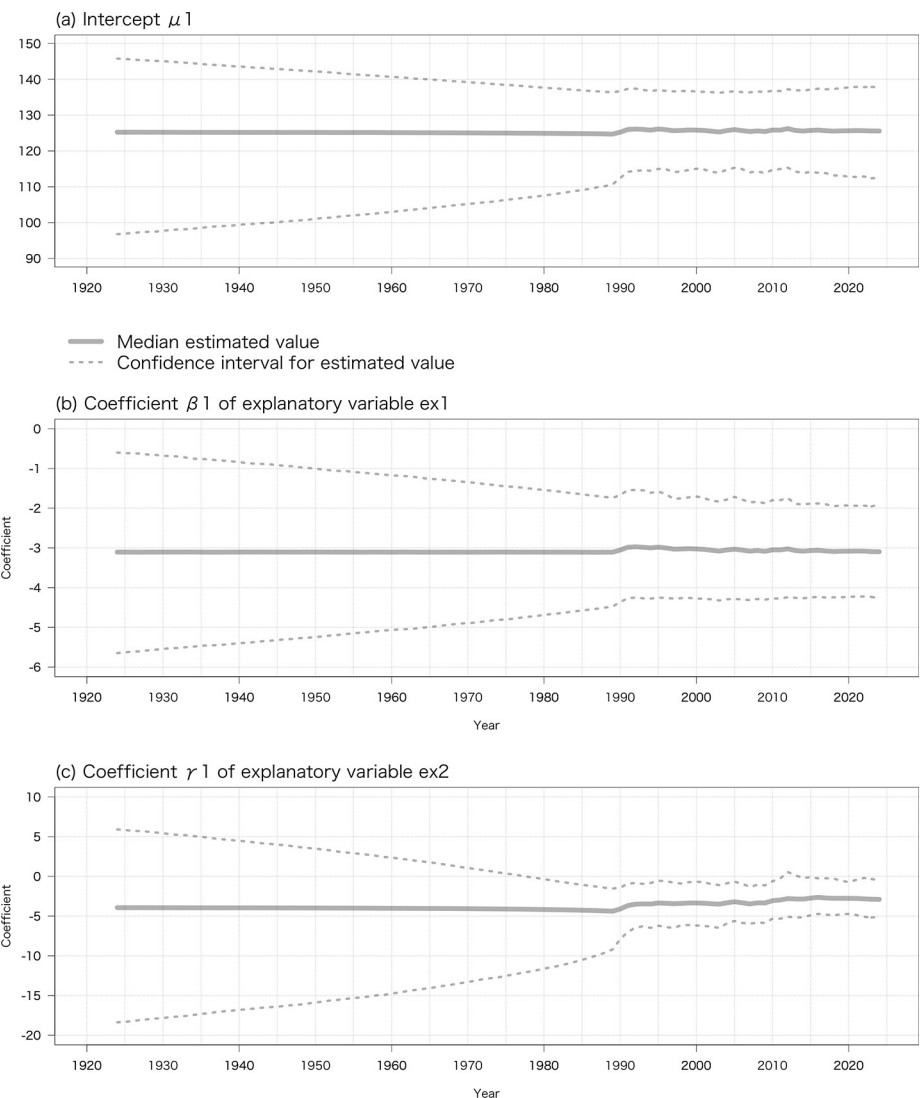

**Fig 3.** Probability distributions of estimated (a) variable μ1$_t$ as the intercept, (b) coefficient β1$_t$ of *ex*1$_t$ as explanatory variable 1, and (c) coefficient γ1$_t$ of *ex*2$_t$ as explanatory variable 2 (μ1$_t$, β1$_t$, and γ1$_t$ in Eqs 1−3) in the case of the estimated model for the true date of FFL.

(Fig 2B), intercept μ1$_t$ (Fig 3A), and coefficients β1$_t$ (Fig 3B) and γ1$_t$ (Fig 3C) estimated only from observed data published on the website since 1989, the width of the 95% credible intervals was narrowed except when there was no observed value of FFB $y$1$_t$.

The mean air temperature on the date of FFL was lower in a climatic region with a low annual mean air temperature than in one with a high annual mean air temperature [44]. This indicates a low cumulative heat requirement for the growth of flower buds in a climatic region where the chilling requirement for release from endodormancy can be met. In addition, in Hachijojima (33°06′44″N, 139°47′01″E), at the southern distribution limit of the full bloom of Yoshino cherry, in years when the chilling requirement for release from endodormancy was not met, the growth of flower buds had a greater heat requirement, and the FFL date tended to be delayed [45]. These facts indicate that the FFL date at a given site in a given year is determined by the balance between the chilling requirement for release from endodormancy and the heat requirement for the growth of flower buds. Therefore, in a region where cherry

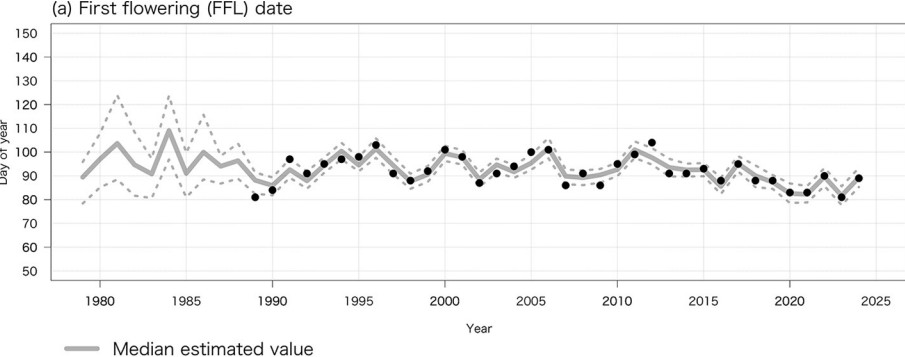

(a) First flowering (FFL) date

— Median estimated value
- - - Confidence interval for estimated value
● Observed data on web site (model input data)

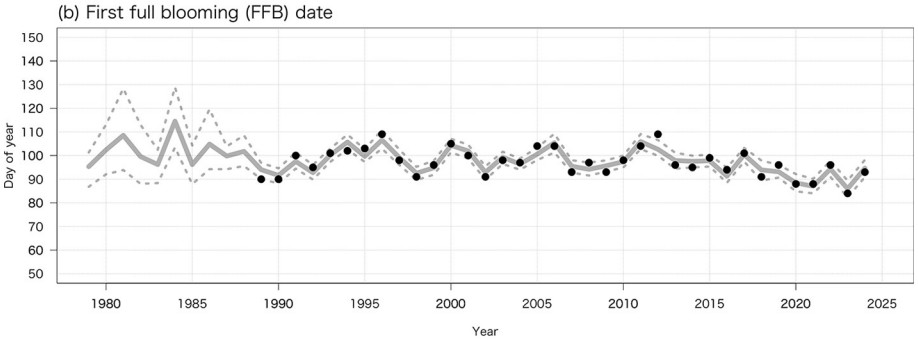

(b) First full blooming (FFB) date

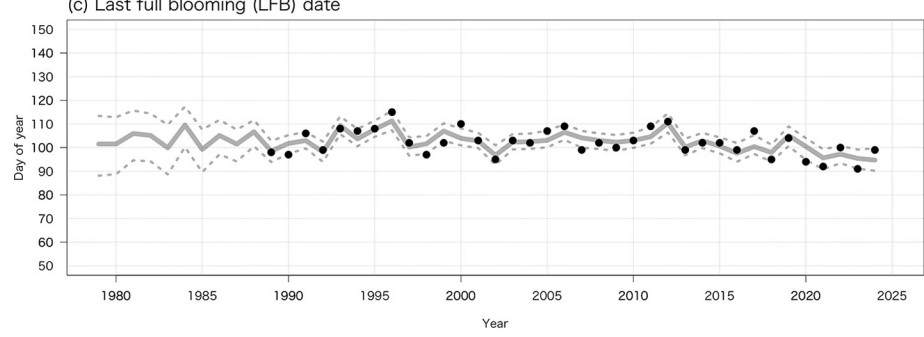

(c) Last full blooming (LFB) date

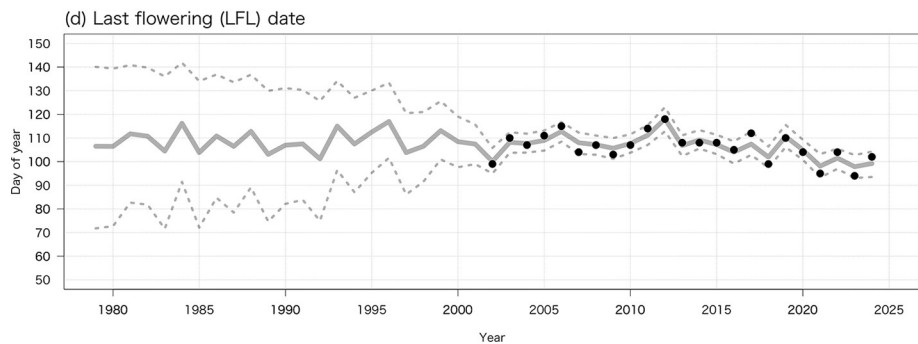

(d) Last flowering (LFL) date

**Fig 4.** Relationship between probability distributions of estimated true dates of (a) FFL, (b) FFB, (c) LFB, and (d) LFL from 1979 to 2024 ($\alpha1_t$ and $\alpha2_t$ in Eqs 4 & 8) by our Bayesian statistical state space model based on temperature data at Tarumi weather station and observed values in records from Motosu City ($y1_t$ and $y2_t$ in Eqs 5 & 9).

**Table 2. Medians of estimated variables μ1$_t$ and μ2$_t$, and coefficients β1$_t$, γ1$_t$, and β2$_t$ based on air temperature data recorded at the Tarumi weather station.**

| Phenology | Parameter | Average | SD |
|---|---|---|---|
| True FFL date | μ1$_t$ | 105.41 | 0.47 |
| | β1$_t$ | −2.84 | 0.04 |
| | γ1$_t$ | −3.47 | 0.62 |
| True FFB date | μ1$_t$ | 112.07 | 0.42 |
| | β1$_t$ | −3.03 | 0.03 |
| | γ1$_t$ | −2.92 | 0.70 |
| True LFB date | μ2$_t$ | 121.68 | 0.69 |
| | β2$_t$ | −2.01 | 0.05 |
| True LFL date | μ2$_t$ | 131.24 | 0.79 |
| | β2$_t$ | −2.48 | 0.08 |

FFL, first flowering; FFB, first full bloom; LFB, last full bloom; LFL, last flowering.

flowering phenology is strongly affected by global warming, the values of the coefficients β1$_t$ and γ1$_t$ (in Eq 4) may change on decadal to centennial time scales. Unlike conventional statistical phenology models which give time-invariant constant coefficients, the proposed Bayesian statistical state space model can evaluate temporal changes in the values of coefficients β1$_t$ and γ1$_t$ (in Eq 4). In Japan, weather stations began modern meteorological observations in the late 19th century [43], and since 1953 have recorded the dates of FFL and FFB of Yoshino cherry by standardized observations [46]. In addition, unstandardized records of the dates of FFL and FFB, which might include uncertainty due to visual inspection, can be traced back to the beginning of the 20th century at several weather stations [5–7]. By applying the proposed Bayesian statistical state space model to these data at multiple points across a wide area at a centennial scale, we can detect the spatiotemporal characteristic of the coefficients β1$_t$ and γ1$_t$ (in Eq 4). This analysis will provide useful evidence for an understanding of the sensitivity and resilience of cherry flowering phenology to climate change.

## Conclusion

We estimated the probability distributions of the year-to-year variability of the true dates of FFL, FFB, LFB, and LFL in Neodani Usuzumi-zakura at a centennial scale, including periods without records of flowering phenology, by applying a Bayesian statistical state space model explained by air temperature data. The estimated values were validated against the retrieved flowering phenology records of the tree from the literature and a private collection. Means of daily minimum air temperature from 1 December to 28/29 February and that of daily mean air temperature from 1 to 31 March explained the true dates of FFL and FFB, while means of daily mean air temperature from 1 to 10 April explained the true dates of LFB and LFL. Results were similar when based on air temperature data recorded at weather stations both 1 km and 29 km from Neodani Usuzumi-zakura, with differences caused by microclimate. These facts indicated that the proposed Bayesian statistical state space model was useful to estimate the true dates of FFL, FFB, LFB, and LFL by using explanatory factors as air temperature with a coarse temporal resolution without the need for labor-intensive calculations for optimization of each parameter in a conventional semi-empirical statistical model. The application of the model to multiple points can successfully estimate the uncertainty of long-term predictions and will deepen our understanding of the spatio-temporal variability of the dates of FFL, FFB, LFB, and LFL at a centennial scale.

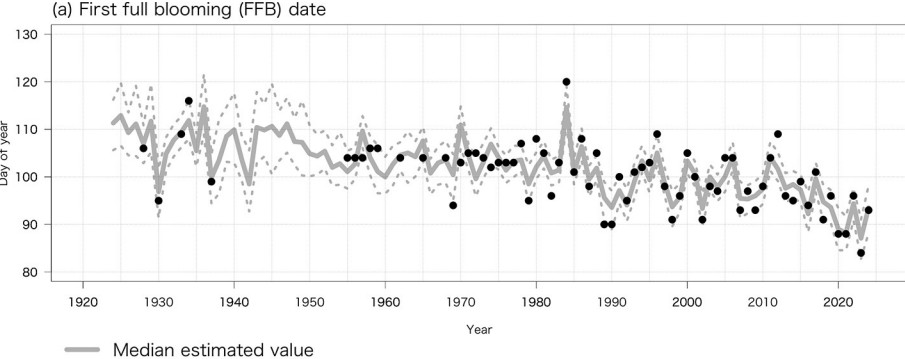

(a) First full blooming (FFB) date

Median estimated value
--- Confidence interval for estimated value
• Observed data (web site + Hakuryu Fujiwara + literature: model input data)

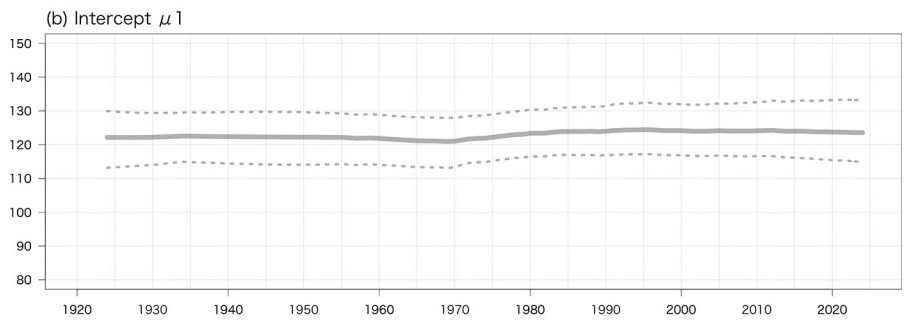

(b) Intercept $\mu 1$

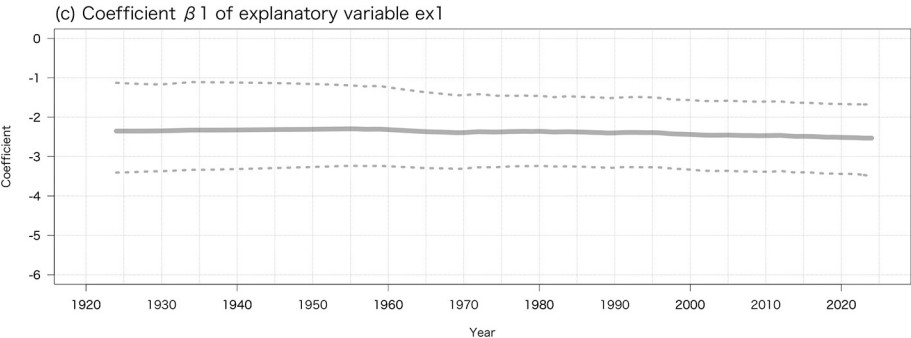

(c) Coefficient $\beta 1$ of explanatory variable ex1

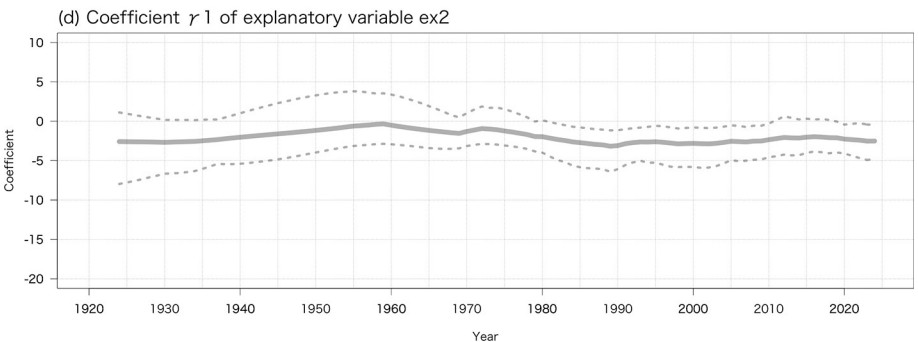

(d) Coefficient $\gamma 1$ of explanatory variable ex2

**Fig 5.** (a) Probability distributions of the estimated true date of FFB from 1924 to 2024 ($\alpha 1_t$ in Eq 4) by the proposed Bayesian statistical state space model based on temperature data at Gifu weather station, and both flowering phenology records of the tree from the literature and a private collection before 1989 (1928–1937: Literature and photograph; 1955–1980: Private collection of Hakuryu Fujiwara; and 1981–1988: Unpublished data of Motosu City [Motosu City {Unpublished}]) and observed data published on a website since 1989 as the observed value of FFB date ($y1_t$ in Eq 5).

Probability distributions of estimated (b) variable $\mu1_t$ as the intercept, (c) coefficient $\beta1_t$ of $ex1_t$ as explanatory variable 1, and (d) coefficient $\gamma1_t$ of $ex2_t$ as explanatory variable 2 ($\mu1_t$, $\beta1_t$, and $\gamma1_t$ in Eqs 1–3) in the case of this estimated model for the true date of FFB.

## Supporting information

**S1 File. Observation records, which were converted to the day of year (DOY) for analysis, in literature, photographs, and the private collection of Hakuryu Fujiwara.**
(CSV)

**S2 File.**
(DOCX)

**S3 File.**
(CSV)

## Acknowledgments

We are grateful to the editors and reviewers for their constructive comments.

## Author Contributions

**Conceptualization:** Nagai Shin.

**Data curation:** Nagai Shin, Hakuryu Fujiwara, Shinjiro Sugiyama.

**Formal analysis:** Nagai Shin, Hiroshi Morimoto.

**Investigation:** Nagai Shin, Hakuryu Fujiwara, Shinjiro Sugiyama.

**Methodology:** Nagai Shin, Hiroshi Morimoto.

**Validation:** Nagai Shin, Hiroshi Morimoto, Taku M. Saitoh.

**Visualization:** Nagai Shin.

**Writing – original draft:** Nagai Shin.

**Writing – review & editing:** Nagai Shin, Hiroshi Morimoto, Taku M. Saitoh.

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
