## [Decision Letter · Decision Letter 0]

19 Nov 2024

PONE-D-24-36692Estimation of true dates of various flowering stages at a centennial scale by applying a Bayesian statistical state space modelPLOS ONE

Dear Dr. Shin,

Thank you for submitting your manuscript to PLOS ONE. After careful consideration, we feel that it has merit but does not fully meet PLOS ONE’s publication criteria as it currently stands. Therefore, we invite you to submit a revised version of the manuscript that addresses the points raised during the review process.

We look forward to receiving your revised manuscript.

Kind regards,

Bahram Heidari

Academic Editor

PLOS ONE

Journal Requirements:

2. Thank you for stating the following financial disclosure: [NS was supported by a KAKENHI grant (JSPS 21H05178) from the Japan Society for the Promotion of Science.

https://www.jsps.go.jp/english/e-grants/index.html

This grant supported our activities regarding data collection and preparation of the manuscript.]. Please state what role the funders took in the study. If the funders had no role, please state: "The funders had no role in study design, data collection and analysis, decision to publish, or preparation of the manuscript." If this statement is not correct you must amend it as needed. Please include this amended Role of Funder statement in your cover letter; we will change the online submission form on your behalf.

3. Thank you for stating the following in the Acknowledgments Section of your manuscript: [This study was supported by a KAKENHI grant (JSPS 21H05178) from the Japan Society for the Promotion of Science.] We note that you have provided funding information that is not currently declared in your Funding Statement. However, funding information should not appear in the Acknowledgments section or other areas of your manuscript. We will only publish funding information present in the Funding Statement section of the online submission form. Please remove any funding-related text from the manuscript and let us know how you would like to update your Funding Statement. Currently, your Funding Statement reads as follows: [NS was supported by a KAKENHI grant (JSPS 21H05178) from the Japan Society for the Promotion of Science.

https://www.jsps.go.jp/english/e-grants/index.html

This grant supported our activities regarding data collection and preparation of the manuscript.]. Please include your amended statements within your cover letter; we will change the online submission form on your behalf.

4. In the online submission form, you indicated that [Insert text from online submission form here]. All PLOS journals now require all data underlying the findings described in their manuscript to be freely available to other researchers, either 1. In a public repository, 2. Within the manuscript itself, or 3. Uploaded as supplementary information. This policy applies to all data except where public deposition would breach compliance with the protocol approved by your research ethics board. If your data cannot be made publicly available for ethical or legal reasons (e.g., public availability would compromise patient privacy), please explain your reasons on resubmission and your exemption request will be escalated for approval.

5. We note that you have referenced (unpublished) on pages 4 and 14, which has currently not yet been accepted for publication. Please remove this from your References and amend this to state in the body of your manuscript: (ie “Bewick et al. [Unpublished]”) as detailed online in our guide for authors

Reviewers' comments:

Reviewer's Responses to Questions

**Comments to the Author**

1. Is the manuscript technically sound, and do the data support the conclusions?

Reviewer #1: Yes

2. Has the statistical analysis been performed appropriately and rigorously? 

Reviewer #1: Yes

3. Have the authors made all data underlying the findings in their manuscript fully available?

Reviewer #1: Yes

4. Is the manuscript presented in an intelligible fashion and written in standard English?

Reviewer #1: Yes

5. Review Comments to the Author

Reviewer #1: This is an excellent empirical study of Bayesian state space modelling. I really enjoyed the detail and the use of novel validation data to demonstrate how this class of modelling can improve the predictive accuracy. Generally the narrative is well written and draws the reader into the problem and the approach. The statistical detail is sufficient for me as a reader with no prior experience with these specific models but who uses other forms of times series data to build predictive models. The results are quite compelling. While the embedded figures are grainy the downloaded .tiff images are excellent and helpful in showing how the validation improves the accuracy. It is an unusual in that is takes it data from phenology data from one single individual tree measure over many decades. The obvious question is why the authors kept the significance of the study merely around the modelling approach rather what the data are used for. I got the sense that the authors were trying to break up the study into one technical description of the methodology (this one) and another (not seen) about the significance of the changes in phenology and what these trends signify in and era where climate change is affection many facets of the life science... especially ecology, evolution and agricultural sciences. I think this is a mistake and missed opportunity. I would like a revision to include how this modelling approach helps us understand the temporal changes in this tree's phenology and what the significance this has for understanding more widespread changes in flower timing under climate change. This obviously has not escaped the authors attention but partitioning the methods and the significance is curious awkward.

6. PLOS authors have the option to publish the peer review history of their article (what does this mean?). If published, this will include your full peer review and any attached files.

Reviewer #1: No

---

## [Author Response · Author response to Decision Letter 0]

17 Dec 2024

Requirement 1: 

Please ensure that your manuscript meets PLOS ONE’s style requirements, including those for file naming. The PLOS ONE style templates can be found at

Answer:

We have checked it.

Requirement 2:

Thank you for stating the following financial disclosure: [NS was supported by a KAKENHI grant (JSPS 21H05178) from the Japan Society for the Promotion of Science.

https://www.jsps.go.jp/english/e-grants/index.html

This grant supported our activities regarding data collection and preparation of the manuscript.]. Please state what role the funders took in the study. If the funders had no role, please state: “The funders had no role in study design, data collection and analysis, decision to publish, or preparation of the manuscript.” If this statement is not correct you must amend it as needed. Please include this amended Role of Funder statement in your cover letter; we will change the online submission form on your behalf.

Answer:

We have removed the “Funding statement section” in the main text.

Requirement 3: 

Thank you for stating the following in the Acknowledgments Section of your manuscript: [This study was supported by a KAKENHI grant (JSPS 21H05178) from the Japan Society for the Promotion of Science.] We note that you have providedfunding information that is not currently declared in your Funding Statement. However, funding information should not appear in the Acknowledgments section or other areas of your manuscript. We will only publish funding information present in the Funding Statement section of the online submission form. Please remove any funding-related text from the manuscript and let us know how you would like to update your Funding Statement. Currently, your Funding Statement reads as follows: [NS was supported by a KAKENHI grant (JSPS 21H05178) from the Japan Society for the Promotion of Science.

https://www.jsps.go.jp/english/e-grants/index.html

This grant supported our activities regarding data collection and preparation of the manuscript.]. Please include your amended statements within your cover letter; we will change the online submission form on your behalf.

Answer:

We have revised the Acknowledgments section as follows. In addition, please see the answer for your second query.

Acknowledgement

We are grateful to the editors and reviewers for their constructive comments.

Requirement 4: 

In the online submission form, you indicated that [Insert text from online submission form here]. All PLOS journals now require all data underlying the findings described in their manuscript to be freely available to other researchers, either 1. In a public repository, 2. Within the manuscript itself, or 3. Uploaded as supplementary information. This policy applies to all data except where public deposition would breach compliance with the protocol approved by your research ethics board. If your data cannot be made publicly available for ethical or legal reasons (e.g., public availability would compromise patient privacy), please explain your reasons on resubmission and your exemption request will be escalated for approval.

Answer:

We want to publish the part of observation records (without the copyright issue). Therefore, we have revised lines 117−120 and updated the supplementary file (csv). 

“Observation records, which were converted to the day of year (DOY) for analysis, in literature, photographs, and the private collection of Hakuryu Fujiwara are listed in the Supplementary file (flowering_phenology_records_Neodani_Usuzumizakura.csv).”

Requirement 5: 

We note that you have referenced (unpublished) on pages 4 and 14, which has currently not yet been accepted for publication. Please remove this from your References and amend this to state in the body of your manuscript: (ie “Bewick et al. [Unpublished]”) as detailed online in our guide for authors

Answer:

We have revised as follows.

Lines in 108−109:

“a photograph in 1930 [33], and unpublished data of Motosu City from 1981 to 1988 (Motosu City, unpublished)”

Lines in 475−476:

“unpublished data of Motosu City [Motosu City {Unpublished}])”

Reviewers’ comments:

Reviewer 1: 

This is an excellent empirical study of Bayesian state space modelling. I really enjoyed the detail and the use of novel validation data to demonstrate how this class of modelling can improve the predictive accuracy. Generally the narrative is well written and draws the reader into the problem and the approach. The statistical detail is sufficient for me as a reader with no prior experience with these specific models but who uses other forms of times series data to build predictive models. The results are quite compelling. While the embedded figures are grainy the downloaded .tiff images are excellent and helpful in showing how the validation improves the accuracy. It is an unusual in that is takes it data from phenology data from one single individual tree measure over many decades. The obvious question is why the authors kept the significance of the study merely around the modelling approach rather what the data are used for. I got the sense that the authors were trying to break up the study into one technical description of the methodology (this one) and another (not seen) about the significance of the changes in phenology and what these trends signify in and era where climate change is affection many facets of the life science... especially ecology, evolution and agricultural sciences. I think this is a mistake and missed opportunity. I would like a revision to include how this modelling approach helps us understand the temporal changes in this tree’s phenology and what the significance this has for understanding more widespread changes in flower timing under climate change. This obviously has not escaped the authors attention but partitioning the methods and the significance is curious awkward.

Answer:

Thank you for your efforts to improve our paper. In accordance with your and editors’ kind and constructive comments, we have revised the manuscript. In addition, we have re-checked all data and revised some minor points and Figs. 2 and 5 (we recalculated Fig. 5). There is no essential problem in these revisions. We will publish the part of observation records (without the copyright issue).

As for your significant comment (you may have clairvoyant abilities), we have revised the third paragraph in the Discussion section as follows (lines 281−303). 

“The mean air temperature on the date of FFL was lower in a climatic region with a low annual mean air temperature than in one with a high annual mean air temperature [44]. This indicates a low cumulative heat requirement for the growth of flower buds in a climatic region where the chilling requirement for release from endodormancy can be met. In addition, in Hachijojima (33°06′44′′N, 139°47′01′′E), at the southern distribution limit of the full bloom of Yoshino cherry, in years when the chilling requirement for release from endodormancy was not met, the growth of flower buds had a greater heat requirement, and the FFL date tended to be delayed [45]. These facts indicate that the FFL date at a given site in a given year is determined by the balance between the chilling requirement for release from endodormancy and the heat requirement for the growth of flower buds. Therefore, in a region where cherry flowering phenology is strongly affected by global warming, the values of the coefficients β1t and γ1t (in Eq. 4) may change on decadal to centennial time scales. Unlike conventional statistical phenology models which give time-invariant constant coefficients, the proposed Bayesian statistical state space model can evaluate temporal changes in the values of coefficients β1t and γ1t (in Eq. 4). In Japan, weather stations began modern meteorological observations in the late 19th century [43], and since 1953 have recorded the dates of FFL and FFB of Yoshino cherry by standardized observations [46]. In addition, unstandardized records of the dates of FFL and FFB, which might include uncertainty due to visual inspection, can be traced back to the beginning of the 20th century at several weather stations [5−7]. By applying the proposed Bayesian statistical state space model to these data at multiple points across a wide area at a centennial scale, we can detect the spatiotemporal characteristic of the coefficients β1t and γ1t (in Eq. 4). This analysis will provide useful evidence for an understanding of the sensitivity and resilience of cherry flowering phenology to climate change.

[45] Shin N, Saitoh TM, Takasu H, Morimoto H. Influence of climate change on flowering phenology of Yoshino cherry at its southern distribution limit. Int J Biometeorol 2024; https://doi.org/10.1007/s00484-024-02797-0”

---

## [Editor Report · Decision Letter 1]

3 Jan 2025

Estimation of true dates of various flowering stages at a centennial scale by applying a Bayesian statistical state space model

PONE-D-24-36692R1

Dear Dr. Shin,

We’re pleased to inform you that your manuscript has been judged scientifically suitable for publication and will be formally accepted for publication once it meets all outstanding technical requirements.

Kind regards,

Bahram Heidari

Academic Editor

PLOS ONE
---

## [Editor Report · Acceptance letter]

14 Jan 2025

PONE-D-24-36692R1 

PLOS ONE

Dear Dr. Shin, 

I'm pleased to inform you that your manuscript has been deemed suitable for publication in PLOS ONE. Congratulations! Your manuscript is now being handed over to our production team.

Kind regards, 

on behalf of

Dr. Bahram Heidari 

Academic Editor

PLOS ONE
